# A New Osteogenic Membrane to Enhance Bone Healing: At the Crossroads between the Periosteum, the Induced Membrane, and the Diamond Concept

**DOI:** 10.3390/bioengineering10020143

**Published:** 2023-01-21

**Authors:** Julie Manon, Robin Evrard, Lies Fievé, Caroline Bouzin, Delphine Magnin, Daela Xhema, Tom Darius, Eliano Bonaccorsi-Riani, Pierre Gianello, Pierre-Louis Docquier, Thomas Schubert, Benoît Lengelé, Catherine Behets, Olivier Cornu

**Affiliations:** 1Morphology Lab (MORF), Institut de Recherche Expérimentale et Clinique (IREC), UCLouvain, 1200 Brussels, Belgium; 2Transplantation and Experimental Surgery Lab (CHEX), IREC, UCLouvain, 1200 Brussels, Belgium; 3Neuromusculoskeletal Lab (NMSK), IREC, UCLouvain, 1200 Brussels, Belgium; 4Centre de Thérapie Cellulaire et Tissulaire Locomoteur, Cliniques Universitaires Saint-Luc, 1200 Brussels, Belgium; 5Department of Orthopaedic and Trauma Surgery, Cliniques Universitaires Saint-Luc, 1200 Brussels, Belgium; 6Imaging Platform (2IP), IREC, UCLouvain, 1200 Brussels, Belgium; 7Bio & Soft Matter (BSMA), Institute of Condensed Matter and Nanosciences (IMCN), UCLouvain, 1348 Louvain-la-Neuve, Belgium; 8Department of Abdominal Surgery and Transplantation, Cliniques Universitaires Saint-Luc, 1200 Brussels, Belgium; 9Department of Plastic and Reconstructive Surgery, Cliniques Universitaires Saint-Luc, UCLouvain, 1200 Brussels, Belgium

**Keywords:** osteogenic membrane, periosteal mesenchymal stem cells, bone allograft, decellularization, cell–matrix interplay, in vitro model for bone regeneration

## Abstract

The lack of viability of massive bone allografts for critical-size bone defect treatment remains a challenge in orthopedic surgery. The literature has reviewed the advantages of a multi-combined treatment with the synergy of an osteoconductive extracellular matrix (ECM), osteogenic stem cells, and growth factors (GFs). Questions are still open about the need for ECM components, the influence of the decellularization process on the latter, the related potential loss of function, and the necessity of using pre-differentiated cells. In order to fill in this gap, a bone allograft surrounded by an osteogenic membrane made of a decellularized collagen matrix from human fascia lata and seeded with periosteal mesenchymal stem cells (PMSCs) was analyzed in terms of de-/recellularization, osteogenic properties, PMSC self-differentiation, and angiogenic potential. While the decellularization processes altered the ECM content differently, the main GF content was decreased in soft tissues but relatively increased in hard bone tissues. The spontaneous osteogenic differentiation was necessarily obtained through contact with a mineralized bone matrix. Trying to deepen the knowledge on the complex matrix–cell interplay could further propel these tissue engineering concepts and lead us to provide the biological elements that allow bone integration in vivo.

## 1. Introduction

Critical-size bone defect (CSBD) has always been a major challenging issue in orthopedic reconstructive surgery. After trauma, infection, oncologic resection, or congenital malformation, the bone architecture is destroyed, and the healing capacity is insufficient due to the defect size, the poor environment, and/or the concomitant damages to the periosteum and surrounding soft tissues. Many researchers have provided some answers for small defects but definite solutions still fail for larger bone gaps over time [1,2,3,4,5]. At present, none of the proposed therapeutic solutions can reach both the important architectural volume to replace and the viability to restore bone homeostasis. To fill in this scientific gap, this study aims to create a novel osteogenic membrane to enhance the integration of massive bone allografts and to deepen the knowledge about the cell–matrix interplay.

In the goal to improve the efficiency of CSBD management, different concepts must be taken into consideration. In actuality, three main entities are known to participate in or massively improve bone consolidation. First, the periosteum is one essential tissue taking part in natural bone repair. This highly-vascularized bilayer collagen membrane covering the bone surface has an evident and unique osteogenic potential for bone healing through endochondral ossification [6,7,8,9]. Thanks to the presence of periosteal mesenchymal stem cells (PMSCs), it contributes to transverse bone growth and bone remodeling, as well as initiating the fracture callus [6,7,8,10,11]. The mesenchymal stem cells’ (MSCs) differentiation towards the osteogenic pathway is regulated by multiple orchestrated signals coming from the surrounding microenvironment, one of the critical ones being the extra-cellular matrix (ECM)–cell interaction [12]. Secondly, the Masquelet technique of the induced membrane also already solicits the biological natural resources of the human body, in the absence of a periosteum, to reproduce a well-vascularized membrane around a cement spacer [13,14,15]. Thirdly, the diamond concept was then described by Giannoudis et al. in order to summarize all the criteria needed to reach nearly 90% of bone regeneration, namely, an osteoconductive matrix, osteogenic cells, osteoinductive mediators, mechanical stability, and vascularization [16].

Based on these concepts, the combination of a bone allograft with a surrounding osteogenic membrane made of a decellularized collagen matrix and seeded with MSC could provide the biological elements for improving bone integration. Previous studies already approached this concept of a multicomponent treatment [17,18,19,20,21,22,23], but some questions still remain unsolved:Which best assembly of grafts should be used? Is there any combination required?How do tissue engineering processes influence the ECM outcomes from a growth factors (GFs) point of view? What happens to their angio- and/or osteoinductive potential?How the spontaneous differentiation of PMSCs towards the osteogenic lineage can be induced without any differentiation culture medium?

This study will attempt to answer these questions by focusing on the fascia lata, PMSCs, and bone allografts. The human fascia lata (HFL), or, more specifically, the iliotibial tract, is a wide tendon sheet extending from the tensor fascia lata muscle to the Gerdy’s tubercle and lying on the vastus lateralis of the quadriceps femoris. It is used for a large panel of therapeutic or reconstructive surgeries [24,25,26,27,28,29,30] and is therefore routinely harvested and processed by our tissue bank [20,31]. Nevertheless, to our knowledge, it has never yet been considered directly for CSBD treatment. This membrane was already compared with the human periosteum (HP) and has proven to be a good candidate to be used as an off-the-shelf membrane scaffold thanks to its wide shapeable surface, its mechanical resistance, and its easy harvesting [32]. However, the absence of osteogenic cells in HFL is critical and needs to be implemented by MSCs. PMSCs are multipotent stem cells deeply confined in the periosteum and were recently shown to play a role in intramembranous ossification as well [33,34]. The superiority of PMSCs over bone marrow MSCs was also previously demonstrated [2,6,35,36,37], in line with their higher osteogenic potential when harvested on load-bearing sites [34].

The association of all these components was tested in this study by assessing the efficacy of de-/recellularization processes on HFL, the PMSC induced/spontaneous differentiation potential, and the presence of osteo-/angiogenic GFs (angio-GFs) in the construct to obtain a function as close as possible to the crossroads between the periosteum and the induced membrane for further in vivo implantation. 

## 2. Materials and Methods

### 2.1. Tissue Harvesting and Processing

#### 2.1.1. Human Fascia Lata (HFL) and Human Periosteum (HP)

HFL tissues were harvested from 5 deceased donors (4 men and 1 woman; average age: 91.3 ± 1.8 years), in the Human Anatomy Department of UCLouvain (IRB00008535, Brussels, Belgium), respecting the local ethics committee authorization (ref. 2021-30AOU-356, approved on the 13th of September 2021).

After a full-length incision of each thigh, going from the anterosuperior iliac spine to the base of the patella, HFL sheets were carefully isolated and cut from the median line to the lateral lip of the linea aspera. Harvested tissues were split into several samples that were differently handled in the following analyses.

The decellularization protocol was the one that was routinely used in our institutional tissue bank, as previously described [20,31]. Briefly, it consists of washing in a NaCl solution and successive baths of pure acetone, ether, ethanol (70°), NaOH (1N), and H_2_O_2_. All of these steps were realized under continuous agitation and separated with a continuous flow of demineralized water. These consecutive detergents ensure sample defatting, cellular membrane digestion, protein coagulation, nuclear acid precipitation, and bacterial, prions, and virus deactivation (such as HIV, HBV, and HCV). Each decellularized ECM was passed through a quality control process for residual chemical reagents, and the pH was adjusted (7.0 < pH < 7.85). The decellularized HFL tissues were then frozen at −80 °C to be stored. The final tissue engineering step was sterilization with gamma-irradiation at min 25 kGy (IBA Mediris, Fleurus, Belgium).

As a comparison, HP tissues were also harvested from the whole femoral circumference, starting from one lip of the linea aspera to the other and between the proximal and distal metaphyses of the same donors. They were processed in exactly the same way as the HFL tissues.

#### 2.1.2. Porcine Periosteal Mesenchymal Stem Cells (PMSCs) and Porcine Bone Allograft (PBA)

Tibial periosteum tissues were harvested from Landrace pigs (n = 10 tibial periostea from N = 5 pigs, young adults of +/− 40 kg) and Aachen mini pigs (n = 4 tibial periostea from N = 2, adults of +/− 40–50 kg). These animals were available after euthanasia from other studies in order to reduce the number of animals (3R concept). All these studies were authorized by the institutional ethics committees (ref. 2020/UCL/MD/027 and A1/UCL/2021-A1, approved on the 14th of September 2020 and the 6th of April 2021, respectively, and ref. 2021/UCL/MD/062, approved on the 5th of November 2021). After a sterile dissection of the lower limb, the periosteum was cautiously removed using a periosteal stripper following a circumferential limit above the distal tibial metaphysis and below the anterior tibial tuberosity. Samples were directly transferred from the operating room to the laminar-flow hood in sterile phosphate-buffered saline (PBS, Sigma-Aldrich, 59321C-1000 mL, Saint Louis, MO, USA) without calcium and magnesium ions. They were then cut into the smallest pieces possible (<1–2 mm), which were immersed in 50 mL Falcons containing a filtered solution of Hanks’ balanced salt solution (HBSS with calcium and magnesium, Gibco, 14025-050) and type 1-A collagenase (0.2%, Sigma-Aldrich, C0130-1G, Saint Louis, MO, USA) and put in a stirring water bath at 37 °C for 3 h. This solution was changed twice an hour while storing the supernatant. The proteolysis was inactivated by adding a two-fold volume of a proliferation medium (PM), namely, the Dulbecco’s Modified Eagle Medium/Nutrient Mixture F-12 (DMEM/F-12, Thermofisher, 11320033, Waltham, MA, USA), and supplemented with fetal bovine serum (FBS, 10%, Gibco, 10270-106), penicillin-streptomycin (1%, Gibco, 15140-122), and amphotericin B (1%, Gibco, 15290-026). All harvested cells were washed twice via successive steps of centrifugation (1500 rpm–5 min) and resuspended in DMEM-F12. Cells were counted before seeding on a 6-well plate (an average of 461,735 cells (+/− 148,014) in each well) and incubated at 37 °C and 5% CO_2_, which was considered the initial passage (P0). The PM was changed every two days and cells were cultured until passage four (P4) in order to select the proliferative PMSCs. PMSCs in P4 were frozen at −80 °C before being analyzed or used.

Porcine forearm bones were harvested from the same previous Landrace pigs with a direct approach between the fasciae of the pectoralis and serratus anterior muscles. The vasculature was carefully isolated and dissected from the axillary vessel to the interosseous artery, until the nutrient artery foramen. The porcine bone allografts (PBA) were harvested with their interosseous pedicles after proximal and distal disarticulation.

### 2.2. Histology

The efficiency of decellularization and recellularization was analyzed using classical histology. For all donors, 1 cm^2^ specimens were embedded in paraffin after fixation in 4% formaldehyde (VWR International, 9713.9010, Radnor, AR, USA). They were then sectioned into 5 μm thick slices and stained with hematoxylin and eosin (H&E), Masson’s trichrome (MT), Picrosirius Red (PSR), and Alcian blue (AB), according to standard protocols. Sections were also stained with 2’-(4-ethoxyphenyl)-5-(4-methyl-1-piperazinyl)-2,5′-bi-1H-benzimidazole trihydrochloride (Hoechst, 2 µg/mL, Life Technologies, H3570, Carlsbad, CA, USA) in order to examine deoxyribonucleic acid (DNA). All sections were digitalized using slide scanners (Leica Microsystems, SCN400, Wetzlar, Germany or a Panoramic ScanII slide scanner, 3DHistech, Budapest, Hungary) with a 20-fold Plan-Apochromat objective or viewed at a 40-fold magnification under a fluorescence microscope (AxioImager Z1, Zeiss, Oberkochen, Germany). The pictures were finally annotated with the open-source Cytomine software (v.3.1.0., www.cytomine.org, first access on the 16th of November 2020) [38] or with SlideViewer software (v2.5.0., Budapest, Hungary), respectively, depending on the slide scanner used.

### 2.3. Immunohistochemistry (IHC)

The same protocols as those previously described [32] were used in order to detect elastin (on tissues), type-1 collagen (col-1), osteocalcin (OC), and runt-related transcription factor 2 (RunX2) (on cell aggregates) on paraffin-embedded slices. Briefly, after paraffin removal, endogenous peroxidase was inactivated with H_2_O_2_ (3%) in methanol. For elastin and col-1, antigen retrieval was achieved via proteinase K (18.7 µg/mL, Roche, 03115828001) incubation in a buffer of tris-ethylenediaminetetraacetic acid (EDTA) (Tris, Merck, 1.08387.250, Darmstadt, Germany; EDTA, Sigma-Aldrich, E5134, Saint Louis, MO, USA) at 37 °C for 20 min. For OC and RunX2, heat-induced epitope retrieval was achieved in a citrate buffer (pH = 5.7) using a microwave. Aspecific binding sites were blocked with 5% bovine serum albumin (BSA, Merck, 12659-500 GM, Darmstadt, Germany) mixed with 0.05% tris-buffered saline (TBS)/Triton (Tris, Merck, 1.08387.250, Darmstadt, Germany; Triton, VWR, M143, Radnor, AR, USA) at RT over 1 h. The slices were incubated with rabbit primary antibodies, i.e., anti-elastin (1:100, Novus biotechne, NB100-2076), anti-collagen I (1:200, Abcam, ab34710), and anti-RunX2 (1:6000, Cell signaling, 12556S), or with mouse primary antibodies, namely, anti-OC (1:200, Thermofisher, MA1-20786, Waltham, MA, USA), at 4 °C overnight. A horseradish peroxidase (HRP)-conjugated antirabbit secondary antibody (Envision, Dako Agilent, K4003, Santa Clara, CA, USA) or an anti-mouse secondary antibody (Envision, Dako Agilent, K4001, Santa Clara, CA, USA) were incubated, respectively, at 4 °C for 1 h. The detection was realized using a 3,3′-diaminobenzidine (DAB) peroxidase substrate (Dako Agilent, K3468, Santa Clara, CA, USA) at RT for 5 min. The counterstaining was achieved using hematoxylin. After mounting with a rapid mounting medium (Entellan new, Sigma-Aldrich, 1.07961.0100, Saint Louis, MO, USA), the slices were also digitalized.

Multiplex immunofluorescence IHC (mIF-IHC) was performed on paraffin-embedded slices based on a published protocol [39]. The same first steps of conventional IHC were performed, including paraffin removal, epitope retrieval, and non-specific antigen blocking with a mix of 5% BSA (Carl Roth, 3854.3, Albumin fraction V, Karlsruhe, Germany) in TBS/Tween20 (0.1%, VWR, 663684B, Radnor, AR, USA) at RT for 30 min. The sections were incubated with the primary antibodies at 4 °C overnight (Table 1). They were washed in TBS/Tween three times. Thereafter, anti-peroxidase-conjugated secondary antibodies were incubated at RT for 45 min and visualized using tyramide signal amplification (TSA) using AlexFluor (AF)-conjugated tyramides in a borate solution (boric acid, Sigma-Aldrich B6768-500gr, Saint Louis, MO, USA; NaCl, VWR, 27810.295-1 kg; H_2_O_2_ 0.003%) at RT for 10 min. After antibody stripping with a new citrate buffer incubation step (this step detached antibodies from tissue sections), the identical protocol was applied with the second primary antibodies, adequate secondary antibodies, and AF-conjugated tyramide amplification following the sequence in Table 1. After a washing step in PBS, nuclei were finally stained with Hoechst (10 µg/mL, Sigma, 14533–100 mg).

After mounting with the Dako fluorescent mounting medium, the sections were digitalized with the fluorescent slide scanner Axioscan.z1 (Zeiss, Oberkochen, Germany), annotated with the AxioVision software (v.4.8.2.0., Zeiss, Oberkochen, Germany), and stored at −20 °C. A negative control, without primary antibodies, was always included to check for absence of detection in both standard IHC and mIF-IHC. The mIF-IHC for RunX2/OC/Hoechst was quantified with the open-source image analysis QuPath software (v0.3.0., University of Edinburgh, Scotland) [40]. Blur areas, damaged tissue, and artefacts were manually excluded. Cells were then detected at a high resolution (0.8 µm/pixel) with a nuclear-based cell classifier relying on the Hoechst staining. Following cell segmentation, cells were classified according to their expression of RunX2 and/or OC. The same parameters were kept constant for all slides. Data were expressed as a percentage of stained cells compared with the total amount of cells labeled via the Hoechst staining.

### 2.4. Scanning Electron Microscopy (SEM)

Scanning electron microscopy (SEM) was used to visualize the presence or removal of cells and the seeding of PMSCs on HFL patches. This technique was the same as previously described [32]. One native and one decellularized sample from each of the four donors were sectioned into 5 mm^2^ samples, fixed on synthetic corks, and immersed in 3% glutaraldehyde. Successive increasing grades of ethanol (30%, 50%, 70%, 80%, 90%, 95%, and 3 × 100%) permitted us to dehydrate the tissues, which were completely dried using the critical point dryer technique (CPD) (Balzers, CPD020). A coating of 10 nm gold (Cressington sputter, 208 HR) created a thin conductive layer on the samples. At least 15 images for each decellularized tissue and donor and a minimum of 30 pictures for the recellularized ECM were generated using field-emission SEM (JSM-7600F, Jeol Ltd. Akishima, Tokyo, Japan) and then examined. The same analysis was performed exactly seven days after PMSC seeding to highlight the superficial recolonization of the ECM.

### 2.5. Extracellular Matrix (ECM) and Cellular Components Quantification

Native (N) and decellularized (D) tissue compositions were compared using the dosages of DNA, collagen, glycosaminoglycan (GAG), and elastin contents. For each quantification, 3 random biopsies were harvested from native and decellularized tissues of 4 donors, resulting in 24 analyzed biopsies (12 native and 12 decellularized) of 20, 25, 10, and 25 mg for collagen, GAG, elastin, and DNA quantification, respectively. The next few steps were exactly reproduced as previously described [32], with the freeze-drying and dry-weighting of all biopsies, DNA extraction using the DNeasy^®^ Blood & Tissue kit (Qiagen, Hilden, Germany), measuring using the Quant-iT PicoGreen DNA assay kit (ThermoFisher, Waltham, MA, USA), quantification of the collagen content in the ECM using the Quickzyme Total Collagen Assay (Quickzyme, Leiden, Netherlands), GAG content extraction and dosage by means of the Blyscan Sulfated-GAG assay kit (Biocolor LTD., Carrickfergus, Northern Ireland), and elastin quantification with the Fast Elastin assay kit (Biocolor LTD., Carrickfergus, Northern Ireland). To increase the precision, plates were filled in duplicate and read three times. After taking the mean of the three readings and the duplicates, the final average quantification of the three biopsies was performed. While the final DNA concentration was expressed in ng/mg of dry weight, ECM protein amounts were expressed in μg/mg of dry weight.

### 2.6. Differentiation through Specific Differentiation Media

PMSCs in P4 were used to evaluate their differentiation capacity into the three phenotypes (chondrogenic, adipogenic, and osteogenic) that characterize MSCs according to the International Society for Cellular Therapy [41,42]. For all 3 differentiations, 1 × 10^5^ cells of each porcine donor were seeded in triplicate in 6-well plates and expanded first with a PM. Specific differentiation media (DM) replaced the PM when 80–90% confluence was reached. All the culture-wells were evaluated by specific colorations on days 7, 14, and 21. A triplicate of the negative control was always performed using PMSCs in classical PM.

#### 2.6.1. Chondrogenic Differentiation—Alcian Blue (AB) Staining

The chondrogenic differentiation was achieved by means of the StemPro™ Chondrogenesis Differentiation kit (Thermofisher, A1007101, Waltham, MA, USA). On day 21, culture-wells were rinsed 3 times in PBS and fixed with 4% formaldehyde at RT for 20 min. After 3 new washings, the AB staining (1%, Sigma-Aldrich, A3157-10G, Saint Louis, MO, USA) was added to the wells for 30 min to stain proteoglycans. The last three washes were performed with HCl (0.1N, Sigma-Aldrich, 1.09057.1000, Saint Louis, MO, USA).

#### 2.6.2. Adipogenic Differentiation—Oil Red O (ORO) Staining

The StemPro™ Adipogenesis Differentiation kit (Thermofisher, A1007001, Waltham, MA, USA) was used following the manufacturer’s protocol. The well preparation with washing and fixation was the same as that for the chondrogenic differentiation assessment. Then, Oil red O (ORO) staining (0.5%, Sigma-Aldrich, O0625-25G, Saint Louis, MO, USA) was added at RT for 10 min to highlight neutral lipids. The last three washes were performed with distilled water.

#### 2.6.3. Osteogenic Differentiation—Alizarin Red (AR) Staining

The osteogenic DM was a mix of classical PM with added dexamethasone (1 µM, Sigma-Aldrich, D4902-25 mg), sodium dihydrophosphate (35 mg/mL, Merck, 1.06346.1000, Darmstadt, Germany), and sodium ascorbate (25 µg/mL, Sigma-Aldrich, A4034-100G, Saint Louis, MO, USA), as already defined by other authors [18]. The Alizarin Red staining (AR, 2%, Sigma-Aldrich, A5533-25G, Saint Louis, MO, USA) followed the same steps of washing in distilled water before and after staining but, for this staining of the calcium deposit, the fixation was in ethanol (70%) at RT for 1 h. AR was incubated at 37 °C for 10 min. After the washes, all wells (chondrogenic, adipogenic, and osteogenic differentiations) were visualized under distilled water with the AxiovertS100 microscope (Zeiss, Oberkochen, Germany) and annotated with the AxioVision software.

#### 2.6.4. Cell Aggregate Creation

Massive cellular gathering spontaneously formed cell aggregates after 12–14 days in the chondrogenic DM and after 14–21 days in the osteogenic DM. Because the osteogenic potential was the primary goal of this study, only osteogenic cell aggregates were characterized through histology and IHC for OC, col-1, and RunX2, as described in Section 2.2 and Section 2.3. Using the same steps as in the standard histology, AR staining (2%, pH 4.2, at 37 °C for 2 min) was also used to highlight the calcium deposit.

### 2.7. Recellularization

In order to evaluate the cytocompatibility of processed scaffolds, cryopreserved PMSCs were seeded for seven days.

#### 2.7.1. PMSC Seeding

Decellularized and sterilized HFL patches (N = 5) of 1 cm² were placed in sterile CellCrown™ inserts (12-well plate inserts, Sigma-Aldrich, Z742383-6EA, Saint Louis, MO, USA) and incubated in classical PM overnight in order to rehydrate and preheat them. The day after, or day zero (D0), 5 × 10^5^ PMSCs were seeded on each patch, and the PM was added after 2 h of dry incubation at 37 °C and 5% CO_2_ to enhance cell adhesion. On day one (D1), the inserts with seeded patches were moved into a new blank 12-well plate to avoid the impact of cells attached to the well walls in future analyses. The PM was replaced every two days just after examining the cell growth (D1, D3, D5, and D7). The cell viability was assessed at the end of the 7th day. The final analyses after seven days also included H&E, MT staining, and mIF-IHC for Col-1/Hoechst, as described above. The whole procedure was performed in duplicate in order to perform a SEM analysis on the whole patches and to assess the cellular spread. All the experimentations included a positive control (cells seeded in wells without a scaffold) and a negative control (scaffold in wells without seeding cells).

#### 2.7.2. Cell Growth—PrestoBlue

The kinetics of cell expansion was assessed using the PrestoBlue Assay (PrestoBlue™ Cell Viability Reagent, ThermoFisher, Invitrogen, A13261, Waltham, MA, USA) on D1, D3, D5, and D7. For this purpose, the PM was replaced by 3 mL of preheated PrestoBlue solution (1:10 reagent with PM) and incubated at 37 °C. After 3 h, 100 µL of the solution was transferred into a 96-well opaque plate, in duplicate, and the fluorescence was measured with a microplate reader (SpectraMax i3x, Molecular Devices; Sunnyvale, CA, USA) at Ex:560/Em:590 nm.

#### 2.7.3. Cell Viability—Live/Dead

After the last PrestoBlue test on D7, Live/Dead staining (LIVE/DEAD™ Viability/Cytotoxicity Kit, ThermoFisher, Invitrogen, L3224, Waltham, MA, USA) evaluated the percentage of cell viability on the whole patch. Wells containing seeded scaffolds were rinsed 3 times with sterile PBS before being submerged in 2 mL of working solution and sheltered from the light for 40 min. After two more washes, patches were transferred onto glass slides and were mounted with Dako fluorescent mounting medium and then visualized using the AxioImager Z1 microscope and processed with the AxioVision software. A total of 5 pictures of each seeded scaffold (at the 4 corners and 1 in the center) were taken at 2.5-fold magnification to average the entire sample area. The cell viability was calculated via the ratio between living cells (green dots) and total cells (living/green + dead/red cells) on pictures using Fiji software (ImageJ-win64, an open-access software on https://imagej.nih.gov/ij/index.html).

### 2.8. Differentiation through ECM

This part of the differentiation experiments aimed to determine if the ECM could initiate osteogenic differentiation on its own, without including the DM. For this purpose, the same seeding as for classical differentiation was performed but, when confluence was reached, instead of feeding the cells with the DM, the ECM was added to the wells, which were supplied only with the PM. AR staining was also performed on D7, D14, and D21. A negative control (only cells with the PM) and a positive one (only cells with the DM) were included and compared with the differentiation through ECM. After the 21st day, the ECM was removed from the culture wells. Residual cells were washed in PBS, scraped out, and centrifugated at 1500 rpm for 5 min. The supernatant was discarded and a pre-warmed agarose solution (3%, VWR, Cambrex Bio Science, 44366 5W-250 g) was poured on the pellet. After solidification in a fridge (15–20 min), agarose blocks were carefully demolded and fixed in 4% formaldehyde to allow mIF-IHC for RunX2/OC. For this last manipulation, a native porcine periosteum (N-PP) was added as an additional tissue control.

#### ECM Preparation

Native and processed porcine bone allografts (N-PBA and D-PBA, respectively), processed HFL (D-HFL), and human periosteum (D-HP) (N = 3 donors for each type of matrices) were used to compare their own osteoinductive potentials. The HFL and HP were processed with the same decellularization protocol as described above. D-PBA was decellularized based on an earlier described perfusion protocol [43,44,45] with successive solutions of sodium dodecyl sulfate (SDS, 1%, VWR, 27926.295, Radnor, AR, USA), Triton X-100 (1%, VWR, M143, Radnor, AR, USA), type-1 DNAse (Roche, Sigma-Aldrich, 11284932001, Saint Louis, MO, USA) at 37 °C and PBS washes. All ECMs were sterilized via gamma-irradiation despite the N-PBA that had been sterilely harvested. All ECMs were rehydrated, and each of them was incubated in a 50 mL Falcon containing the PM for 24 h before being added into the culture wells. This permitted us to maximize the recovery of the respective GF in case the ECM had already released them into the medium. These preloaded PMs were retrieved and used to change the mediums every two days in the respective culture wells. 

### 2.9. Growth Factors (GFs)—Immunoblot

Two main types of GF were explored: the angiogenic and the osteogenic. Angio-GFs present and/or preserved after the tissue engineering process would considerably increase the potential in vivo revascularization after transplantation as well as cell survival or new cellular attraction. Bone morphogenetic proteins (BMPs) are GFs that exert a relevant osteogenic function. Immunoblots allowed us to appraise the intrinsic angiogenic/osteogenic potentials of all these GFs present in our different types of ECM. For these quantifications, native periosteum (N-HP) and fascia lata (N-HFL) tissues (N = 3, also for both native tissues) were added to compare and evaluate the persistence of GFs after decellularization.

#### 2.9.1. Protein Extraction from ECM

The protein extraction differed between tissue types. Indeed, hard bone tissues could not be dissolved using soft tissue techniques. For soft tissues (N/D-HP and N/D-HFL), a 50 mg biopsy was chopped and lysed with a radioimmunoprecipitation assay (RIPA) buffer containing a protease inhibitor cocktail and Pho-Stop. A Precellys Homogenizer (Bertin Technologies SAS, Montigny-le-Bretonneux, France) melded the samples in 4 consecutive cycles at 7200 rpm. The samples were centrifugated at 2000 rpm for 5 min, and only the clear residual supernatant was collected.

For their part, hard bone tissues (N/D-PBA) were ground with an electric saw and pestle and mortar to obtain 500 mg biopsies of bone powder. The protein extraction was then achieved with an extraction solution made of 100 mL of guanidine HCl (8 M, Sigma-Aldrich, G7294-100ML, Saint Louis, MO, USA), 10 mL of N-ethylmaleimide (0.2 M, Sigma-Aldrich, E3876-25G, Saint Louis, MO, USA), 5 mL of benzamidine (0.2 M, Sigma-Aldrich, B6506-25G, Saint Louis, MO, USA), 2 mL of phenylmethylsulphonyl (0.2 M, Sigma-Aldrich, P7626-1G), and 83 mL of demineralized water under continuous agitation at 4 °C for 24 h. A Tris HCl buffer (50 mM, Trizma^®^ hydrochloride solution, Sigma-Aldrich, T2663-1L, Saint Louis, MO, USA) was added to the supernatant and incubated on a shaking plate at 4 °C for 5 h. The samples were centrifugated at 3000 rpm for 10 min and only the residual supernatant was collected. 

For all extractions, the protein concentration in the homogenized supernatant was defined using the PierceTM BCA Protein assay kit (ThermoFisher, 23227, Waltham, MA, USA), following the manufacturer’s protocol.

#### 2.9.2. Angiogenic GF Quantification

The RayBio^®^ C-Series Human Angiogenesis Antibody (Tebu-Bio, 126AAH-ANG-1-8, Le Perray-en-Yvelines, France) quantified 20 different angio-GFs. For this purpose, membranes were blocked at RT for 30 min and incubated with 100 μg of proteins on a shaker plate at RT for 2.5 h. Next, the steps exactly followed the manufacturer’s protocol, with a few washes, incubations in a biotinylated antibody cocktail, and dilution in HRP-Streptavidin, as well as baths of detection buffers preceding the chemiluminescent revelation on chromatographic films. Based on a published method [11] and using Fiji software, each densitometry spot was subtracted from the background density and normalized to the ratio of the positive control of an arbitrarily chosen reference tissue (N-HP) to the positive control of the analysis in question. The normalization was completed via the multiplication of the obtained value by the global quantity of proteins to the weight of each respective sample (expressed in μg of proteins/mg of tissue). Final outcomes were averaged on the mean density of two detection spots for each GF and the mean of all three donors.

#### 2.9.3. Bone Morphogenetic Protein (BMP) Quantification

Exactly the same procedure was followed in order to quantify BMP-2, -4, -5, -6, -7, -8, -9, and -11 using the RayBio^®^ C-Series Human BMP Related Array 2 (Tebu-Bio, 126AAH-BMP-2–8, Le Perray-en-Yvelines, France).

### 2.10. Statistical Analyses

Statistical analyses were realized using GraphPad Prism (v.8.0.1, GraphPad Software, San Diego, CA, USA) as well as SPSS software (v.27, IBM SPSS, Inc., Chicago, IL, USA). A Shapiro–Wilk test and Q–Q plots checked the normality of continuous variables. All data with a Gaussian distribution were compared using parametric unpaired T-tests for DNA and ECM components comparisons before/after decellularization, as well as for the Live/Dead. Levene’s test was always used to check the equality of variances beforehand. A Welch’s correction was applied if the latter was not assumed. Kruskal–Wallis tests were used to analyze angiogenic or osteogenic GF content, followed by post hoc multiple comparisons (Mann–Whitney) if justified and adjusted with a Bonferroni correction. All tests were two-tailed. The level of significance was always set at 0.05 in order to reject the null hypothesis. All outcomes displayed in the graphics are expressed as means and the standard error of the means (SEMs).

## 3. Results

### 3.1. Decellularization

Macro- and microscopic aspects illustrated the decellularization success on 1124 µm (±108 μm) thick HFL samples since all of Crapo’s criteria were met [46]. Both the H&E and Hoescht staining showed a no-man’s-land concerning nuclei (Figure 1a), and the DNA content fell below the critical threshold of 50 ng/mg of dry weight (Figure 1b, *p* = 0.000008). The ECM was also preserved according to the qualitative and quantitative persistence of collagen (*p* = 0.323), which is the main ECM component of the HFL [32]. GAG and elastin were qualitatively preserved but quantitatively altered via the decellularization process (*p* < 0.0001).

### 3.2. PMSC Differentiation and Cell Aggregates

PMSCs from Landraces and mini pigs behaved the same way and showed differentiation into three different phenotypes on day 21 (Figure 2a). The PMSCs from only one Landrace failed to differentiate into the chondrogenic phenotype but were positive for the adipogenic and osteogenic ones. Among all wells (7 donors x triplicate = 21 wells for each phenotype), 5 osteogenic cell aggregates were spontaneously formed between the 14th and 21st days. Their characterization (Figure 2b) showed an aggregation of cells of different shapes on the H&E-stained sections, expressing RunX2 and osteocalcin, and surrounded by a new ECM containing col-1 fibers. The calcium deposit was highlighted via the AR staining, attesting to the osteogenic phenotype of these cell aggregates.

### 3.3. Recellularization and Cytocompatibility

The cytocompatibility of D-HFL was confirmed. Indeed, after PMSC seeding, all the histological stainings and the mIF-IHC showed cell recolonization on the surface with the onset of invasion deeper into the tissue (Figure 3a). The col-1 fibers were organized to support the cell bed in recellularized tissue. The SEM supported that the scaffold was a welcoming environment for the PMSCs, as the latter were numerous and well spread out, which is a sign of well-being. The Live/Dead staining pointed in the same direction, with qualitatively very few red/dead cells visualized compared with the green/alive ones. Quantitatively, the PMSC growth followed the same kinetic growth curve as the positive control wells, and both viabilities were close to 100% on the 7th day (*p* = 0.190) (Figure 3b).

### 3.4. Angio-Induction Potentials of ECMs

Angio-GFs were studied with the aim of measuring the angio-induction potentials of different ECMs after transplantation. First, our scaffold of interest (N/D-HFL) was compared with N/D-HP, which is richly vascularized in order to approximate the baseline for both soft tissues (Figure 4a). The main angio-GFs present in N-HP were β-FGF, IGF-1, and VEGF-D, whereas N-HFL contained more angiogenin and TIMP-2. However, after decellularization, both tissues lost a certain amount of all angio-GFs, but the best preserved one was IGF-1 in both tissues. In order to evaluate and/or to choose the right complex association of ECM, the profile graph and the heatmap (Figure 4b) allow us to illustrate the relatively higher quantity of all angio-GFs for D-PBA. IGF-1 was always the most abundant, except for N-PBA. Each angio-GF was statistically compared between all the ECMs (Table 2).

### 3.5. Osteoinduction Potentials of ECMs

#### 3.5.1. BMP Content within the ECMs

When comparing D-HFL with the native one or N/D-HP, which is also the innate natural osteogenic membrane, we observed that N-HP contained much more BMP than N-HFL (Figure 5a). However, after decellularization, D-HFL kept more BMP than D-HP. The heatmap and the profile graph (Figure 5b) both indicate that BMP-2 and BMP-7 were the most abundantly expressed BMPs in all the ECMs. However, there were statistical differences between each ECM depending on the studied BMP (BMP-2, -5, -7, or -11) with a higher content found in D-PBA (Table 3).

#### 3.5.2. Spontaneous In Vitro Differentiation

PMSCs clearly differentiated in contact with D-PBA and N-PBA in the same manner as with the DM (Figure 5c). At a higher magnification (×20), some sparse PMSCs in the D-HFL wells could suggest the beginning of a calcium deposit even though no clear differentiation was visible. The same observation was made for the D-HP wells but to a lesser extent. After 21 days of culture, mIF-IHC showed some positive osteogenic markers in almost all wells but in different proportions. The control N-PP confirmed the OC staining of the remaining bone harvested together with the periosteum, but no periosteal cell expressed either OC or RunX2. According to the quantification, an interesting observation is that the ECMs producing a clear osteodifferentiation with calcium deposits in the AR staining have the highest proportions of RunX2-/OC+ and/or RunX2+/OC+ cells (Figure 5d).

## 4. Discussion

The actual gold standard for hard tissue grafting is the bone autograft, which integrates most criteria of the diamond concept. Nevertheless, its quantitative availability is limited and only suitable for small defects [5,47]. The harvest site morbidity is also a point of consideration [47,48,49,50]. All these reasons usually direct CSBD treatment towards bone allografts, which were considered in this study. Because of their burden of complications due to the lack of viability [1,3], the need for a synergic association was evident in order to complete the diamond.

A new bone component (cancellous or morselized bone graft, massive allograft, or other approaches) wrapped by a membrane has already been described as relevant so as to prevent soft tissue interposition and ectopic ossification as well as to guide osteogenesis [17,51]. While the periosteum is the natural shell of the bone, the induced membrane could reproduce its main function of supplying the biological needs to promote bone regeneration. Even if very helpful for CSBD treatment by ensuring the function of a vascularized periosteum-like tissue, this induced membrane requires a two-stage surgery with the associated disadvantages and differs from HP in its composition, thickness, and mechanical isotropy [14]. In actuality, the induced membrane is thicker and contains more MSCs than HP [52]. The goal of this study was not to reproduce these tissues identically, but rather to use the knowledge of these membranes to learn how to mimic their function. The creation of an off-the-shelf induced membrane that could ensure the function sought in a one-stage surgery may emerge as a solution to reduce the time, costs, and drawbacks for society and patients involved in this challenging surgical treatment.

Although the HFL was already used for a huge panel of indications in the reconstruction surgery field, it was never used to repair CSBD. In this study, this musculoskeletal tissue was, however, a very good cell carrier for PMSCs. A recent study also demonstrated that PMSC viability, survival, and migration were better when seeded on a collagen membrane than when in a synthetic bone [53]. The complex multicomponent association still seems mandatory, because even if PMSCs brought the osteogenicity, contact with a mineralized matrix was needed to provide the osteoinduction. All these outcomes allowed us to answer our previous questions, but some points deserve to be argued.

First, the chondrogenic and osteogenic potential of PMSCs alone in vitro, without any ECM interaction, was promising in itself. A recent study also showed the bipotentiality of PMSCs, which can help in endochondral and intramembranous ossification [9,36]. The spontaneous formation of cell aggregates expressing RunX2, producing a col-1-based matrix, and generating calcium deposits was very encouraging for the purpose of bone regeneration. 

Furthermore, during the natural bone healing steps, the initial fracture bleeding gives rise to a hematoma, which provides a series of cytokines, GFs (VEGF, IGF, FGF, TGF-β, EGF, PDGF, CCL5, as well as BMP), and pro-inflammatory factors that promote initial healing [9,16,36]. All these molecules work in the mitogenesis process, osteoblastic differentiation, as well as neovascularization to restore the appropriate environment for soft/hard callus formation. 

The tissue engineering processes have already been largely explored, and the decrease in GAG and elastin in different connective tissues was previously described in the literature [12,43,44,46,54] but did not affect the goal of obtaining a malleable and consistent membrane, because the HFL is mostly composed of col-1 fibers [32]. However, the GAG depletion and/or dysfunction after decellularization were hypothesized to influence the GF linkage to the ECM [12,46,54].

For example, the remaining GAG from decellularized tissues would not be able to fix FGF and TGF-β1 [55] or have an effect on VEGF binding [56]. IL-8 (CXCL8) is also directly attached to sulfated GAG chains [56]. In addition to their vascular function, these angio-GFs also play a function in bone homeostasis and/or osseous regeneration, i.e., IGF-1, FGF, and TGF-β activate osteoblastic proliferation, increase col-1 synthesis, and reduce osteoblast apoptosis [57,58,59]. An interesting observation in our study was that despite a distinct angio-GF profile in N-HP and N-HFL, the decellularization process modified the matrices to obtain a strictly similar profile between both tissues with the highest preservation for IGF-1, which intervenes at both the osteogenic and angiogenic levels [60]. The specific roles of IGF-1 include osteogenesis induction and the formation of endothelial cells both by the phosphoinositide-3-kinase (PI3-K)/Akt and the mitogen-activated protein kinase (MAPK) pathways as well as the control of bone homeostasis by regulating the RANKL [60,61]. Therefore, in terms of angiogenic potential, D-HP did not show superiority over D-HFL. Angio-GFs present and/or preserved after the tissue engineering process would increase the potential in vivo revascularization after transplantation as well as cell survival or new cellular attraction. While the angiogenic potentials of the different ECMs were explored in this study, the angiogenesis could further be evaluated using a coculture of PMSCs and endothelial cells.

From an osteogenic point of view, GAG seemed to have a significant impact on the biological activities of BMPs and improve the expression of RunX2, a key pro-osteogenesis regulator gene, as well as the subsequent osteoblastic differentiation [56]. The tissue engineering processes did not affect the BMP content of our ECMs in the same way. Indeed, the total BMP amount was much more lowered in D-HP than in D-HFL. Since N-HP is much more cellular than N-HFL [32], GF proteins are produced and secreted in higher amounts, and the absolute GF content of the intracellular compartment probably explained the higher total quantity of BMP in N-HP. On the contrary, the BMPs expressed in N-HFL are probably already fixed to the ECM because of its fewer cells and are then more resistant to cellular decimation. The relatively higher residual BMP content in D-HFL is in favor of using it as a cell carrier with an osteogenic purpose. This does not necessarily mean that D-HFL could start osteogenic differentiation. Nevertheless, some authors recently indicated that fibroblasts lying on the HFL displayed a weak yet real osteogenic differentiation capacity [62], suggesting that this ECM might preserve a few osteoinduction potentials for MSCs. Even though a few cells showed some calcium metabolism, PMSCs started to express osteogenic markers as soon as they were placed into culture wells with the PM, as highlighted with the mIF-IHC. This fact is very encouraging because PMSCs are probably the mesenchymal stem cells pre-programmed to preferentially differentiate towards the osteogenic pathway. However, this cellular behavior was clearly not enough to induce positive AR staining. The explanation may lie in the percentage of labeled cells and the degree of associated differentiation. Indeed, RunX2 is a very early nuclear gene marker in the osteogenic differentiation pathway (osteoprogenitor marker), while OC secretion labels occur at a later stage of differentiation (osteoblast marker) [34]. The very last stage of osteogenic differentiation still expresses OC but loses RunX2 expression. Only mineralized ECMs (N/D-PBA) were able to increase the percentage of advanced stages of differentiation, i.e., RunX2+/OC+ and RunX2-/OC+ cells, as much as needed to allow spontaneous osteogenic differentiation. Previous authors showed that the demineralization of a scaffold plays a relevant role in reducing the in vitro osteogenic differentiation of bone marrow MSCs compared with a mineralized scaffold [63]. Interestingly, in complete opposition to the soft tissue, hard bone tissues benefited from a more important GF expression after decellularization. This difference may be related to the organization of the mineralized bone matrix that hides GFs in N-PBA and that may be unmasked via decellularization. D-PBA has been shown to contain significantly more BMP-2 and -7, which are some of the most relevant factors for osteoinduction [64,65,66]. They can initiate fracture healing and play a crucial role in osteoblast differentiation [64,66] as well as in mature bone homeostasis but sometimes lead to ectopic ossifications [16,47,66]. To avoid this side effect, taking advantage of the BMPs that are naturally active in D-PBA and/or containing them via a periosteal flap [67], for example, could orientate and contain their action. The induced increase in those BMPs’ expressions in D-PBA could potentially compensate for the loss in D-HFL, send an osteogenic signal, and initiate PMSC differentiation while avoiding the complications of external BMP [65,68].

This study aimed at characterizing the engineered D-HFL and its interaction with PMSCs, in comparison with the HP, so as to produce the best osteogenic complex to maximize the impact on in vivo models. The perfect multicomponent device to guarantee bone regeneration could then associate a D-HFL as a carrier of PMSCs seeded on its internal side, surrounding a D-PBA that could serve as an osteoconductive matrix as well as an osteoinductive signal for PMSC differentiation after transplantation. Keeping this in mind, undifferentiated PMSCs could be used in order to maintain their proliferative and replicative potential while ensuring progressive osteogenic differentiation in vivo in contact with D-PBA. In contrast, if the decellularization could avoid the immune response after allotransplantation [69] and improve BMP expression in D-PBA, it might lead to a drop in the acute inflammatory response responsible for the initiation of bone healing. Moreover, surrounding the bone with a tubing sheet could create a hypoxic chamber that might contain the surgical hematoma and the inflammatory process initiating the healing procedure. This new question rising from this in vitro study could only be addressed using in vivo models. As has already been developed in our institution [70], a femoral non-union pig model will assess the new osteogenic membrane in stringent conditions of a CSBD typical of the clinical situation. This model will be representative of what orthopedic surgeons face in patients and will allow, in the case of success, us to consider the first clinical trials of this innovative approach.

## 5. Conclusions

Our data allow us to consider the association between a bone allograft, which can fill in the missing volume (stability) while providing osteogenic GFs (osteoinduction), and a surrounding D-HFL membrane (osteoconduction), shapeable on demand, containing some angio-GFs (vascularization) and seeded with PMSCs (osteogenic cells). This complex in vitro model could provide the biological elements that allow bone regeneration in vivo.

## Figures and Tables

**Figure 1 bioengineering-10-00143-f001:**
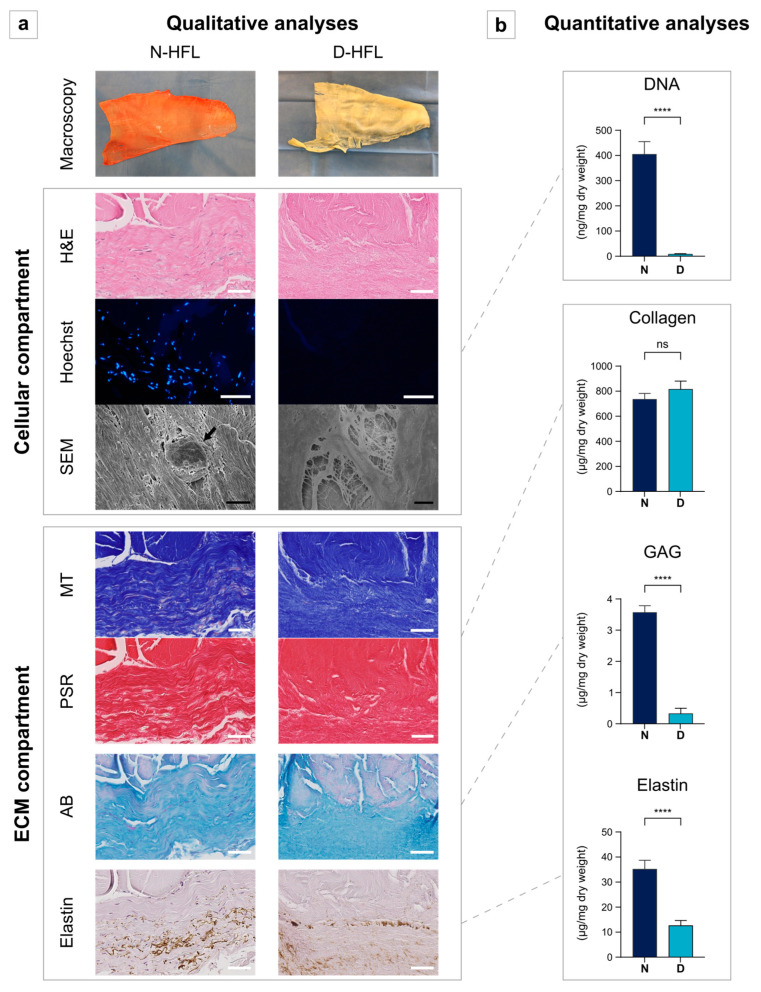
Qualitative (**a**) and quantitative (**b**) evaluation of the cellular and extracellular matrix compartments before/after decellularization. The DNA extraction was achieved with the DNeasy^®^ Blood & Tissue kit (Qiagen, Italy) and measured using the Quant-iT PicoGreen DNA assay kit (ThermoFisher Scientific). The quantification of the collagen content was carried out using the Quickzyme Total Collagen Assay (Quickzyme, Leiden, Netherlands), the GAG content extraction and dosage by means of the Blyscan Sulfated-GAG assay kit (Bio-color LTD., Carrickfergus, Northern Ireland), and the elastin quantification with the Fast Elastin assay kit (Biocolor LTD., Carrickfergus, Northern Ireland). ECM: extracellular matrix, N-HFL: native human fascia lata, D-HFL: decellularized fascia lata, H&E: hematoxylin and eosin, SEM: scanning electron microscopy, MT: Masson’s trichrome showing collagen fibers, PSR: Picrosirius Red showing collagen fibers, AB: Alcian blue showing proteoglycans, DNA: deoxyribonucleic acid, and GAG: glycosaminoglycans. Black arrow shows a cell. White scale bars: 50 µm. Black scale bars: 5 µm. ****: *p* < 0.0001; ns: non-significant.

**Figure 2 bioengineering-10-00143-f002:**
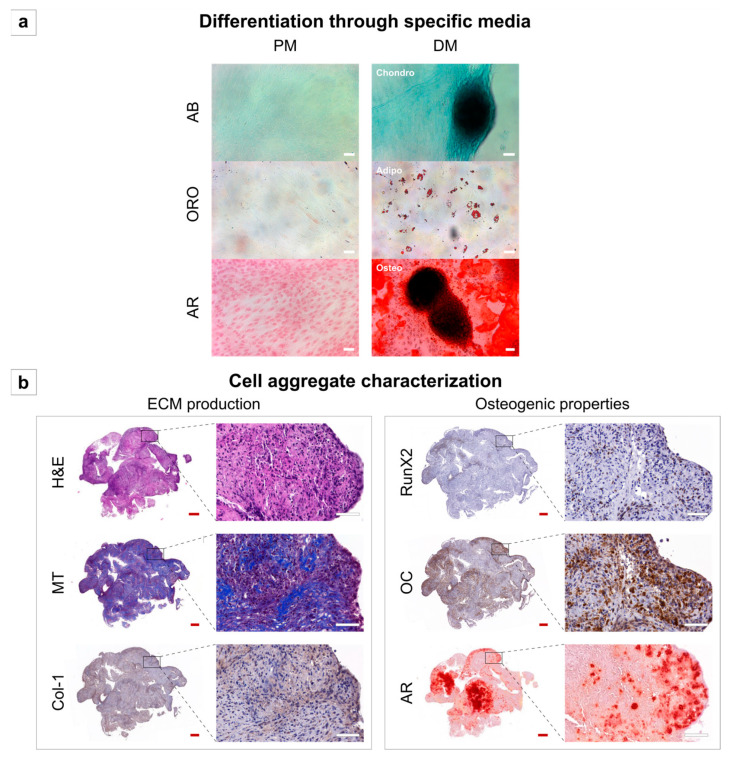
Porcine periosteal mesenchymal stem cell differentiation through chondrogenic, adipogenic, and osteogenic differentiation media (**a**) and cell aggregate characterization after spontaneous formation between 14th and 21st days (**b**). All figures were taken on the 21st day. PM: proliferation medium, DM: differentiation medium, Chondro: chondrogenic differentiation medium, Adipo: adipogenic differentiation medium, Osteo: osteogenic differentiation medium, AB: Alcian blue staining, ORO: Oil red O staining, AR: Alizarin Red staining, H&E: hematoxylin and eosin staining, MT: Masson’s trichrome staining, Col-1: type-1 collagen immunohistochemistry (IHC), and OC: osteocalcin IHC. White scale bars: 50 µm. Red scale bars: 200 µm.

**Figure 3 bioengineering-10-00143-f003:**
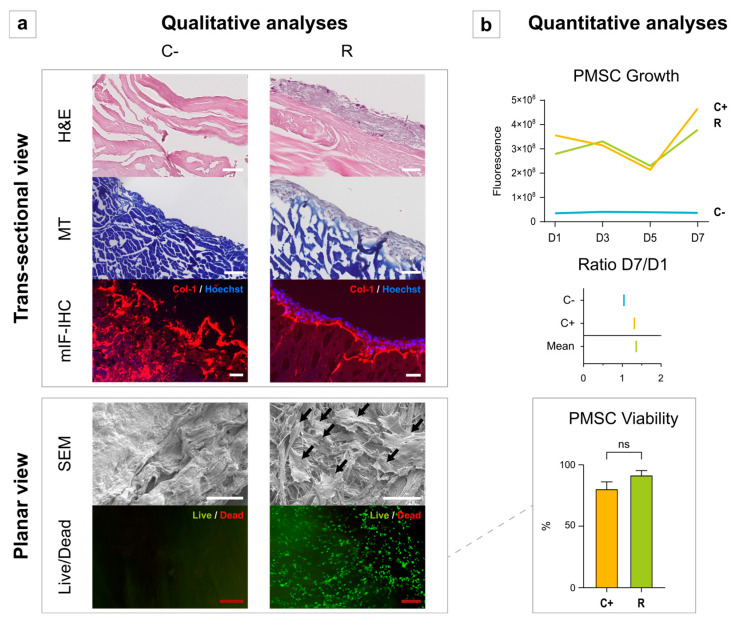
Qualitative (**a**) and quantitative (**b**) evaluation of D-HFL scaffold recellularization with PMSCs. The PMSC growth was assessed using the PrestoBlue Assay (PrestoBlue™ Cell Viability Reagent, ThermoFisher, Invitrogen, A13261, Waltham, MA, USA) on days (D) 1, 3, 5, and 7. The ratio D7/D1 expresses the ratio of the number of cells on D7 to that on D1. The Live/Dead staining (LIVE/DEAD™ Viability/Cytotoxicity Kit, ThermoFisher, Invitrogen, L3224, Waltham, MA, USA) evaluated the cell viability by the ratio between the number of living cells and the total number of cells on the patch. C-: negative control (scaffold without cells), R: recellularized scaffold, C+: positive control (cells without scaffold), H&E: hematoxylin and eosin, MT: Masson’s trichrome, mIF-IHC: multiplex immunofluorescence immunohistochemistry, Col-1: type-1 collagen, SEM: scanning electron microscopy, and Dx: day number x. Black arrows show seeded PMSCs. White scale bars: 50 µm. Red scale bars: 200 µm. ns: non-significant.

**Figure 4 bioengineering-10-00143-f004:**
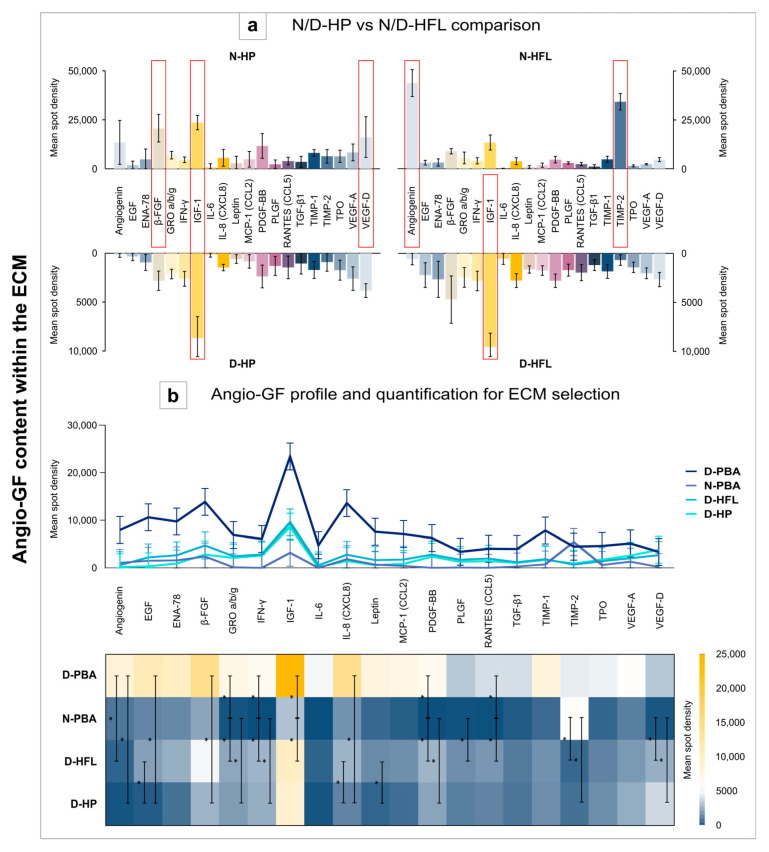
Evaluation of angio-induction potentials of ECMs via the angiogenic GF content in each matrix. The RayBio^®^ C-Series Human Angiogenesis Antibody (Tebu-Bio, 126AAH-ANG-1-8, Le Perray-en-Yvelines, France) was used for this quantification (**a**,**b**). First, the D-HFL membrane, which is our membrane of interest, was compared with the periosteum (**a**). Then, the angiogenic GF content was expressed according to each ECM with a profile graph and a heatmap (**b**). Angio-GF: angiogenic growth factor, N/D-HP: native or decellularized human periosteum, N/D-HFL: native or decellularized human fascia lata, D-PBA: decellularized porcine bone allograft, and N-PBA: native porcine bone allograft. Error bars: 95% confidence interval (CI); *: *p* < 0.05.

**Figure 5 bioengineering-10-00143-f005:**
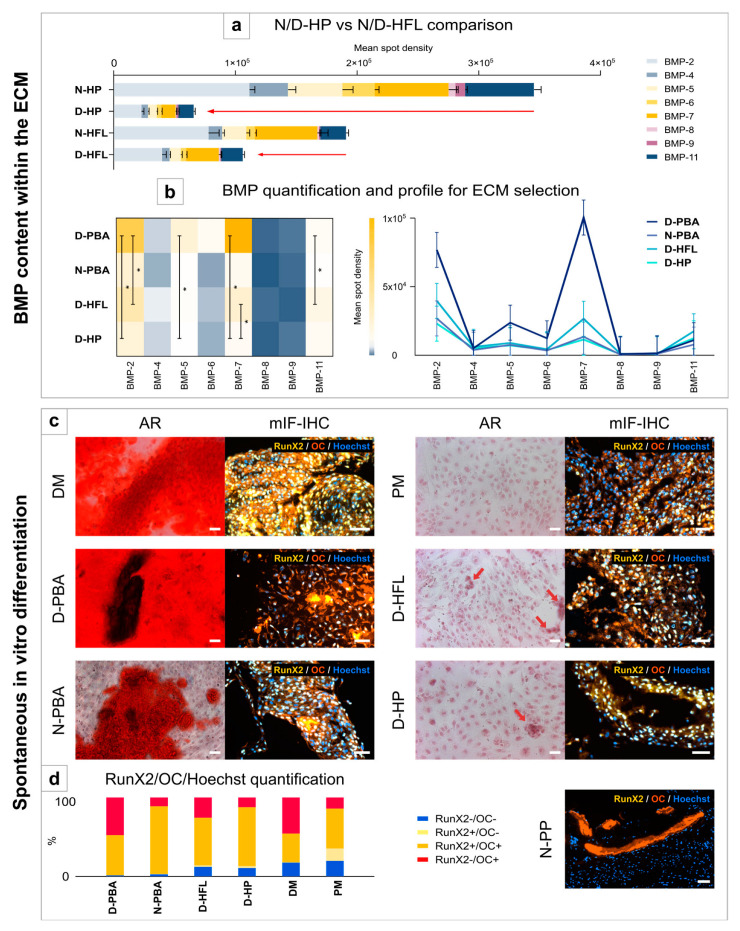
Evaluation of the osteoinduction potentials of ECMs via the BMP content in each matrix and spontaneous differentiation of PMSCs in vitro. The RayBio^®^ C-Series Human BMP Related Array 2 (Tebu-Bio, 126AAH-BMP-2–8, Le Perray-en-Yvelines, France) was used for the BMP quantification (**a**,**b**). First, the D-HFL membrane was compared with the periosteum via a stacked bars diagram of all types of BMP (**a**). Then, the BMP content was expressed according to each ECM with a heatmap and a profile graph (**b**). PMSC spontaneous differentiation was confirmed with AR staining and mIF-IHC for RunX2 and OC (**c**). This last experiment was quantified in order to increase the knowledge about the ECM impact on PMSC differentiation stage (**d**). This graph shows the relative content of cells labeled with RunX2 alone (RunX2+/OC-), RunX2 and OC (RunX2+/OC+), OC alone (RunX2-/OC+), or none of them (RunX2-/OC-). N/D-HP: native or decellularized human periosteum, N/D-HFL: native or decellularized human fascia lata, BMP: bone morphogenetic protein, D-PBA: decellularized porcine bone allograft, N-PBA: native porcine bone allograft, AR: Alizarin Red staining, mIF-IHC: multiplex immunofluorescence immunohistochemistry, OC: osteocalcin, DM: differentiation medium, PM: proliferation medium, and N-PP: native porcine periosteum. White scale bars: 50 µm. T-bars: standard error of the mean (SEM), error bars: 95% confidence interval (CI), and *: *p* < 0.05.

**Table 1 bioengineering-10-00143-t001:** Combination of primary and secondary antibodies, and AlexaFluor-conjugated tyramides for multiplex immunofluorescence IHC.

#	Antigen	Primary Antibody			Secondary Antibody	AF-Conjugated Tyramide		
		Dilution	Company	Ref.	Polymer HRP	Company	Ref.	Fluorochrome
1	Col-1	1:500	Abcam	ab34710	Anti-rabbit	InvitroGen	B40955	AF555
1	RunX2	1:6000	Cell signaling	12556S	Anti-rabbit	InvitroGen	B40955	AF555
2	OC	1:200	Thermofisher	MA1-20786	Anti-mouse	InvitroGen	B40958	AF647

#: Sequence of use, Ref: Catalog number, and AF: AlexaFluor.

**Table 2 bioengineering-10-00143-t002:** Comparison of angiogenic growth factor contents between each different ECM.

	Kruskal–Wallis			Mann–Whitney Post Hoc		
Dependent Variable	df	H	*p*-Value	Tissue 1	Tissue 2	*p*-Value
Angiogenin	3	15.30	0.002	D-PBA	D-HP	0.012 ^a^
					D-HFL	0.024 ^a^
EGF	3	16.78	0.001	D-PBA	D-HP	0.012 ^a^
				D-HFL	D-HP	0.012 ^a^
ENA-78	3	10.36	0.016			
β-FGF	3	11.53	0.009	D-PBA	D-HP	0.024 ^a^
GRO a/b/g	3	13.85	0.003	N-PBA	D-HP	0.012 ^a^
					D-HFL	0.012 ^a^
					D-PBA	0.012 ^a^
IFN-γ	3	13.57	0.004	N-PBA	D-HP	0.012 ^a^
					D-HFL	0.012 ^a^
					D-PBA	0.012 ^a^
IGF-1	3	16.29	0.001	N-PBA	D-HFL	0.012 ^a^
					D-PBA	0.012 ^a^
IL-6	3	9.30	0.026			
IL-8 (CXCL8)	3	13.10	0.004	D-PBA	D-HP	0.024 ^a^
				D-HFL	D-HP	0.012 ^a^
Leptin	3	12.19	0.007	D-HFL	D-HP	0.012 ^a^
MCP-1 (CCL2)	3	10.75	0.013			
PDGF-BB	3	13.97	0.003	N-PBA	D-HP	0.012 ^a^
					D-HFL	0.012 ^a^
					D-PBA	0.012 ^a^
PLGF	3	9.16	0.027	N-PBA	D-HFL	0.012 ^a^
RANTES (CCL5)	3	11.82	0.008	N-PBA	D-HFL	0.012 ^a^
					D-PBA	0.012 ^a^
TGF-β1	3	6.53	0.093			
TIMP-1	3	10.17	0.017			
TIMP-2	3	8.65	0.034	N-PBA	D-HP	0.012 ^a^
					D-HFL	0.012 ^a^
TPO	3	5.61	0.132			
VEGF-A	3	4.13	0.248			
VEGF-D	3	12.95	0.005	N-PBA	D-HP	0.012 ^a^
					D-HFL	0.012 ^a^

^a^ With Bonferroni correction for multiple comparisons. Kruskal–Wallis tests were used in order to analyze angiogenic GF content and were followed by post hoc multiple comparisons (Mann–Whitney) if justified and adjusted with a Bonferroni correction. All tests were two-tailed. EGF: epidermal growth factor, ENA-78: epithelial neutrophil-activating peptide 78, β-FGF: beta fibroblast growth factor, GRO a/b/g: growth-regulated oncogene a/b/g, IFN-γ: interferon-gamma, IGF-1: insulin-like growth factor 1, IL-6: interleukin 6, IL-8 (CXCL8): interleukin 8 (C-X-C motif ligand 8), MCP-1 (CCL2): monocyte chemoattractant protein 1 (C-C motif ligand 2), PDGF-BB: platelet-derived growth factor BB monomer, PLGF: placental growth factor, RANTES (CCL5): regulated upon activation normal T cell expressed and secreted (C-C chemokine ligand 5), TGF-β1: transforming growth factor-β1, TIMP-1 and -2: tissue inhibitors of metalloproteinase 1 and 2, TPO: thrombopoietin, VEGF-A and D: vascular endothelial growth factor A and B, N-PBA: native porcine bone allograft, D-PBA: decellularized porcine bone allograft, D-HP: decellularized human periosteum, D-HFL: decellularized human fascia lata, df: degrees of freedom, and H: the test statistic of Kruskal–Wallis test. Grayed-out boxes highlight angio-GFs with a significant difference in Kruskal–Wallis tests= and for which a post hoc multiple analysis was performed.

**Table 3 bioengineering-10-00143-t003:** Comparison of BMP content between each different ECM.

	Kruskal–Wallis			Mann–Whitney Post Hoc		
Dependent Variable	df	H	*p*-Value	Tissue 1	Tissue 2	*p*-Value
BMP-2	3	13.25	0.004	D-PBA	D-HFL	0.012 ^a^
					D-HP	0.012 ^a^
BMP-4	3	3.07	0.381			
BMP-5	3	10.79	0.013	D-PBA	D-HP	0.012 ^a^
BMP-6	3	3.33	0.343			
BMP-7	3	14.37	0.002	D-PBA	D-HP	0.012 ^a^
				D-HFL	D-HP	0.012 ^a^
BMP-8	3	4.51	0.211			
BMP-9	3	5.78	0.123			
BMP-11	3	11.97	0.007	D-PBA	D-HFL	0.012 ^a^

^a^ With Bonferroni correction for multiple comparisons. Kruskal–Wallis tests were used in order to analyze angiogenic GF content and were followed by post hoc multiple comparisons (Mann–Whitney) if justified and adjusted with a Bonferroni correction. All tests were two-tailed. BMP: bone morphogenetic protein, N-PBA: native porcine bone allograft, D-PBA: decellularized porcine bone allograft, D-HP: decellularized human periosteum, D-HFL: decellularized human fascia lata, df: degrees of freedom, and H: the test statistic of Kruskal–Wallis test. Grayed-out boxes highlight angio-GF with a significant difference in Kruskal–Wallis test and for which a post hoc multiple analysis was performed.

## Data Availability

All data analyzed during the study are included in this published article. The complete original datasets generated during the current study are available from the corresponding author on reasonable request.

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
