# Peer review of "A New Osteogenic Membrane to Enhance Bone Healing: At the Crossroads between the Periosteum, the Induced Membrane, and the Diamond Concept"

_bioengineering, 2023, doi:10.3390/bioengineering10020143_

Round 1

Reviewer 1 Report

A manuscript entitled ‘A New Osteogenic Membrane to Enhance Bone Healing: at the Crossroads Between The Periosteum, the Induced Membrane 3 and the Diamond Concept’ is a careful and clinically-relevant study, which is well presented (notably, their re-cellularization results) and easy to follow. I have only a few comments for the authors to address.

Major comments

1.       Please indicate HFL thickness

2.       Please indicate cell seeding density for initial PBMC isolation, passaging and differentiation assays

3.       Please provide more details on HP harvesting (anatomical areas, sizes, harvesting technique and ideally, any images before and after de-cell)

4.       Similarly, provide more information of PBA processing (size of the graft, processing technique)

5.       In the Introduction the authors state that “PMSC are multipotent stem cells deeply confined in the cambial layer” – this is not entirely correct as PMSCs were recently described in the fibrous layer (https://doi.org/10.1155/2019/6074245)

6.       The authors state that IM “differs from HP by its composition, thickness, and mechanical isotropy”. They should also mention common features as described in https://doi.org/10.1016/j.bone.2013.08.009.

7.       The authors are aiming to develop a one-stage procedure to critical-size bone defect repair. But in Conclusions, they refer to seeding the membrane with PMSCs. Would PMSCs need to be expanded (as in the present study)? That way, this would require a two-stage procedure unless allogeneic PMSCs are used. How are these allogeneic PMSCs harvested, expanded and banked? The authors should better explain their vision.

  Minor comments

1.       Consider changing from “spontaneously formed spheroids” to “spontaneously formed cell aggregates”. The word ‘spheroids’ commonly refers to controlled cell assembly whereas in your case, the process was not specifically controlled/induced. Most likely the aggregates were formed due to uneven differentiation or matrix deposition across the well.

2.       Line 335, “ECM was added to the wells while keeping the supply in PM” – please clarify/re-write.

3.       Provide more discussion on the specific role of IGF-1 in angiogenesis promotion for bone repair.

Author Response

Dear Reviewer,

On behalf of all the authors, I would like to thank you for the time and effort you dedicated to reviewing our manuscript titled “A new osteogenic membrane to enhance bone healing : At the crossroads between the periosteum, the induced membrane and the diamond concept” as well as for your pertinent comments and suggestions. This in-depth criticism has enabled us to significantly improve our manuscript. Please find below our point by point answers to your comments. Your initial feedback, copy-pasted in grey italic, is directly followed by the corresponding answer, written in normal black text.

Reviewer #1

              Yes        Can be improved            Must be improved          Not applicable

Does the introduction provide sufficient background and include all relevant references?

              (x)          ( )          ( )           ( )

Are all the cited references relevant to the research?

              ( )           (x)         ( )           ( )

Is the research design appropriate?

              (x)          ( )          ( )           ( )

Are the methods adequately described?

              (x)          ( )          ( )           ( )

Are the results clearly presented?

              (x)          ( )          ( )           ( )

Are the conclusions supported by the results?

              (x)          ( )          ( )           ( )

A manuscript entitled ‘A New Osteogenic Membrane to Enhance Bone Healing: at the Crossroads Between The Periosteum, the Induced Membrane and the Diamond Concept’ is a careful and clinically-relevant study, which is well presented (notably, their re-cellularization results) and easy to follow. I have only a few comments for the authors to address.

Major comments

  1. Please indicate HFL thickness

The HFL thickness (mean +/- standard error of the mean) was added in the results. By the way, this measure was already discussed more in detail in our previous article : Manon J, Evrard R, Maistriaux L, Fievé L, Heller U, Magnin D, Boisson J, Kadlub N, Schubert T, Lengelé B, Behets C and Cornu O (2022), Periosteum and fascia lata: Are they so different? Front. Bioeng. Biotechnol. 10:944828. doi: 10.3389/fbioe.2022.944828.

  1. Please indicate cell seeding density for initial PBMC isolation, passaging and differentiation assays

For the differentiation assays, cell density was already mentioned in the appropriated section of material and methods (M&M) (1x105 cells for the point 2.6. Differentiation through specific differentiation media as well as for the point 2.8. Differentiation through ECM).

For the initial isolation, it depended on the cell count after the collagenase solution. The choice of the number of wells to be seeded depended on this cell count. Mostly, average of 461,735 cells (+/- 148,014) were seeded in each well. This value was added in the text.

Concerning the passaging, the guideline is to pass the cells when the well reaches 80-90% of confluence. This guideline was applied. However, the cell count was always checked after each passage just to insure the good growth curve of the cells. All these details are listed in a separate excel sheet that we can provide if necessary.

  1. Please provide more details on HP harvesting (anatomical areas, sizes, harvesting technique and ideally, any images before and after de-cell)

Recommended details were added in the appropriate M&M section. We did not add the macroscopy view of the HP since it could burden an existing figure or the entire manuscript by adding an additional figure which we are not convinced about its added value.

  1. Similarly, provide more information of PBA processing (size of the graft, processing technique)

The harvesting technique was already described in the manuscript in the section 2.1.2. and the processing technique was also described in the section 2.8.1. Since the whole section of M&M is still quite heavy, we mentioned 2 previous papers explaining the details of the decellularization protocol and we also added one more paper, under publication, which describe in-depth all the details needed for PBA processing.

  1. In the Introduction the authors state that “PMSC are multipotent stem cells deeply confined in the cambial layer” – this is not entirely correct as PMSCs were recently described in the fibrous layer (https://doi.org/10.1155/2019/6074245)

Thank you for your sharing, we modified the statement by erasing the cambial layer precision. Also, the citation was added to support the superiority of PMSC on BMSC.

  1. The authors state that IM “differs from HP by its composition, thickness, and mechanical isotropy”. They should also mention common features as described in https://doi.org/10.1016/j.bone.2013.08.009.

It is also a really interesting study and we are delighted to add its reference.

  1. The authors are aiming to develop a one-stage procedure to critical-size bone defect repair. But in Conclusions, they refer to seeding the membrane with PMSCs. Would PMSCs need to be expanded (as in the present study)? That way, this would require a two-stage procedure unless allogeneic PMSCs are used. How are these allogeneic PMSCs harvested, expanded and banked? The authors should better explain their vision.

It is a very good question. Actually, the use of allogenic or autogenic PMSC also remains a debate in the literature but also for us. As you know, the autogenic cells transplantation showed the best results by mean of function and immunocompatibilty. But it would require a first surgical procedure to harvest those autologous cells and expand them. The patient will then sustain 2 different surgeries. But, PMSC could also be harvested in an ectopic site, far from the sick zone, leading to a “one-stage” surgery in the injured area (for example, you could harvest PMSC from the antero-medial side of the tibia to further repair a CSBD of the femoral shaft). However, it would take time to expand the cells and to be ready to do the reconstructive surgery. In the other hand, it will probably give better results for young patients than for a 80-years-old-patient with a massive decrease of the amount and function of PMSC. For this reason, the allogenic transplantation is not excluded and must be considered because it could be harvested on common procedure when the periosteum is thrown, like during an osteotomy in children correction (we could harvest the periosteum on the thrown bone edge). PMSC could then be isolated, expanded and stored exactly in the same conditions as described in our paper. When reaching P4, the cells could be stored in a predefined quantity at -80°C fridges, already present in our tissue bank, and could be further thawed and use in an off-the-shelf manner. It could be also a solution to avoid function discrepancy between different donors like some previous studies that already tried to gather several harvestings to get a homogeneous and reproductible function. This point of view is interesting but it also sustains the disadvantages of allogenic transplantation such as immunogenicity which is not yet completely understood in the field of bone regeneration. As you can see, this topic is a singular debate and could be discussed deeper and longer why we chose not to address in detail in our manuscript. However, it is an unavoidable reflection that will have to be part of the future in vivo studies.

  Minor comments

  1. Consider changing from “spontaneously formed spheroids” to “spontaneously formed cell aggregates”. The word ‘spheroids’ commonly refers to controlled cell assembly whereas in your case, the process was not specifically controlled/induced. Most likely the aggregates were formed due to uneven differentiation or matrix deposition across the well.

It has been changed in the whole manuscript as well as figure 2.

  1. Line 335, “ECM was added to the wells while keeping the supply in PM” – please clarify/re-write.

It has been changed in the main text.

  1. Provide more discussion on the specific role of IGF-1 in angiogenesis promotion for bone repair.

It has been provided.

We are very grateful for your comments and queries, which were quite thoughtful and in-depth for us. Therefore, we have adapted the manuscript accordingly, thereby improving the paper’s content, as requested. We modified the main text and manuscript by adding the previously explained modifications marked up using the “Track Changes” function as requested.

Once again, thank you so much for allowing us to clarify the open issues.

Sincerely Yours,

The authors.

Reviewer 2 Report

The article is really well written and present a novel idea which would be interesting to see applied in in vivo studies.

Author Response

Dear Reviewer,

On behalf of all the authors, I would like to thank you for the time you dedicated to reviewing our manuscript titled “A new osteogenic membrane to enhance bone healing : At the crossroads between the periosteum, the induced membrane and the diamond concept”.

Reviewer #2

Yes        Can be improved            Must be improved          Not applicable

Does the introduction provide sufficient background and include all relevant references?

              (x)          ( )          ( )           ( )

Are all the cited references relevant to the research?

              (x)          ( )          ( )           ( )

Is the research design appropriate?

              (x)          ( )          ( )           ( )

Are the methods adequately described?

              (x)          ( )          ( )           ( )

Are the results clearly presented?

              (x)          ( )          ( )           ( )

Are the conclusions supported by the results?

              (x)          ( )          ( )           ( )

The article is really well written and present a novel idea which would be interesting to see applied in in vivo studies.

We are very grateful for your comment, thank you so much.

Sincerely Yours,

The authors.

Reviewer 3 Report

The study aims to create a novel osteogenic membrane to enhance the integration of massive bone allograft. However, the current research content were not in a close relationship with the problem the author intended to solve, and the experimental design and research content were insufficient. It was suggested to revise the research purpose according to the actual problems to be solved. In addition, more relavant experientments were sugguested to be added.

1. In the Abstract part, the structure is not well-organized. And the purpose and main results are unclear. It would be better to briefly state the research background and purpose first, following the summary of materials & methods, the main research results, and conclusions.

2. Please perform a semi-quantitative analysis of the fluorescence of coll-1 in Fig.3. From the picture, the overall collagen seems to decrease after PMSC loading, please explain.

3. In the Introduction part, the arthors said “This study will attempt to answer these questions by focusing on fascia lata, PMSC and bone allograft.

 • Which best assembly of grafts should be used ? Is there any combination required ?

• How tissue engineering processes influence the ECM outcomes from a growth factors (GF) point of view ? What happens to their angio- and/or osteo-inductive potential ?

• Do MSC have to be used/transplanted already differentiated or not ?

However, from the view of experimental design and the obtained results, this paper is not enough to answer the above three questions. Overall, the main research contents of this paper focused on exploring the influence of liberalization process on the angio-and/or osteo-inductive potential of human Facia Lata (HFL). More relavant experientments were sugguested to be conducted:

1)Cell-loaded osteo-inductive potential-related experiments were lacking, which would help to demonstrate the need for combination cells.

2)“Do MSC have to be used/transplanted already differentiated or not ?” No relavant experiments have been conduted to demonstrate the problem.

3)The authors only tested the angiogenic growth factors content after the decellularization of HFL, it is suggested to add the observation of angiogenesis after coculture with angiogenic cells in vitro.

4. The author aims to create a novel osteogenic membrane used in critical bone defects to enhance the integration of massive bone graft . However, the manuscript does not reflect the advantages of its application. It is recommended to establish animal models for better verification.

5. The English expression is not very clear, it is suggested to modify the language.

Author Response

Dear Reviewer,

On behalf of all the authors, I would like to thank you for the time and effort you dedicated to reviewing our manuscript titled “A new osteogenic membrane to enhance bone healing : At the crossroads between the periosteum, the induced membrane and the diamond concept” as well as for your pertinent comments and suggestions. This in-depth criticism has enabled us to significantly improve our manuscript. Please find below our point by point answers to your comments. Your initial feedback, copy-pasted in grey italic, is directly followed by the corresponding answer, written in normal black text.

Reviewer #3

              Yes        Can be improved            Must be improved          Not applicable

Does the introduction provide sufficient background and include all relevant references?

              ( )           (x)         ( )           ( )

Are all the cited references relevant to the research?

              (x)          ( )          ( )           ( )

Is the research design appropriate?

              ( )           ( )          (x)          ( )

Are the methods adequately described?

              ( )           (x)         ( )           ( )

Are the results clearly presented?

              ( )           (x)         ( )           ( )

Are the conclusions supported by the results?

              ( )           ( )          (x)          ( )

The study aims to create a novel osteogenic membrane to enhance the integration of massive bone allograft. However, the current research content were not in a close relationship with the problem the author intended to solve, and the experimental design and research content were insufficient. It was suggested to revise the research purpose according to the actual problems to be solved. In addition, more relavant experientments were sugguested to be added.

  1. In the Abstract part, the structure is not well-organized. And the purpose and main results are unclear. It would be better to briefly state the research background and purpose first, following the summary of materials & methods, the main research results, and conclusions.

We understand your comment. Our abstract contains all the points you mentioned, and we would prefer to structure it with subtitles to make it clearer. Nevertheless, the writing of the abstract followed scrupulously the structure recommended by the guidelines for authors of the journal.

  1. Please perform a semi-quantitative analysis of the fluorescence of coll-1 in Fig.3. From the picture, the overall collagen seems to decrease after PMSC loading, please explain.

There is no specific explanation for a potential decrease of coll-1 since the most deleterious process was the decellularization. The coll-1 amount was quantified in this purpose. However, concerning the recellularization, it is a very interesting comment because the goal of this picture was to illustrate the reorganization after the seeding, with specific markers for the cells as well as for the ECM separately. The main observation was centered on the high density of cellular colonization on the patch surface. An additional examination led us to consider the better reorganization of the collagen fibers just under the cells bed so as to support them as briefly explained in the manuscript. The heterogeneity of the center of the matrix could come from the variation of the cutting axis in the different tissues. Indeed, collagen-1 fibers are better highlighted when the tissue is cut alongside with the main axis of these fibers. After consultation with some other authors, we did not think that a semi-quantitative analysis would add value to this manuscript and to the questions we are trying to solve. The membrane needs to be a good scaffold to support cells growth, proliferation and differentiation and those statements are demonstrated. Moreover, in this kind of analysis and as explained in the M&M section (point 2.7.1.), we used to repeat five times the “experimental patch” but they are usually compared to one well-known and previously described negative control.

  1. In the Introduction part, the arthors said “This study will attempt to answer these questions by focusing on fascia lata, PMSC and bone allograft.”

“ • Which best assembly of grafts should be used ? Is there any combination required ?

  • How tissue engineering processes influence the ECM outcomes from a growth factors (GF) point of view ? What happens to their angio- and/or osteo-inductive potential ?
  • Do MSC have to be used/transplanted already differentiated or not ?”

However, from the view of experimental design and the obtained results, this paper is not enough to answer the above three questions. Overall, the main research contents of this paper focused on exploring the influence of liberalization process on the angio-and/or osteo-inductive potential of human Facia Lata (HFL).

Thank you for your comment and viewpoint. From our point of view, we brought some answers to all of these 3 questions considering all our results. Also, the discussion was structured around those questions and stated this conclusion according the point of interest of this article (N/D-HFL – N/D-PBA – Un-/differenciated PMSC) : “The perfect multicomponent device to guarantee bone regeneration could then associate a D-HFL as a carrier of PMSC seeded on its internal side, surrounding a D-PBA which could serve as osteoconductive matrix as well as osteoinductive signal for PMSC differentiation after transplantation. Keeping this in mind, undifferentiated PMSC could be employed in order to maintain their proliferative and replicative potential while ensuring progressive osteogenic differentiation in vivo in contact with D-PBA.

Indeed, this kind of study could be enlarged to all possible membrane scaffolds, to all possible hard scaffolds and to all possible stem cells niches but it was not the precise purpose of our work. Our work was led by some perspectives that we described a little bit more below.

More relavant experientments were sugguested to be conducted:

1)Cell-loaded osteo-inductive potential-related experiments were lacking, which would help to demonstrate the need for combination cells.

In our manuscript, the spontaneous in vitro differentiation experiment allowed to explore the osteo-inductive potential of the combination of PMSC and ECM in the same culture well. The angiogenic potential was not practically experimented in our paper but we think that it was not your question. By the way, the angiogenic potential in vitro, after PMSC and/or endothelial cells (co-)culture could be an interesting perspective.

If your question is about in vivo osteo-induction, of course, this will be the purpose of further studies.

2)Do MSC have to be used/transplanted already differentiated or not ?” No relavant experiments have been conduted to demonstrate the problem.

In this study, we evaluated the capacity of undifferentiated PMSC to spontaneously differentiate towards the osteogenic pathway in contact with specific ECM (which could be the best combination to transplant). As you correctly mentioned, we did not undergo experiment with already differentiated PMSC (if you consider that the positive control with the differentiation medium was not an experiment on cells whose differentiation was induced) but actually, it was not needed to answer our question. Most of the previous literature used to use already differentiated cells to transplant in a patient. However, there are few disadvantages of using differentiated cells ; you need to add 3 or 4 weeks of differentiation time before being ready for the surgery and you lose the replicative potential of stem cells. The purpose of our study was to demonstrate that we could use undifferentiated cells in order “to maintain their proliferative and replicative potential while ensuring progressive osteogenic differentiation in vivo in contact with D-PBA” which answer the question. The use of undifferentiated PMSC would also facilitate further clinical translations.

3)The authors only tested the angiogenic growth factors content after the decellularization of HFL, it is suggested to add the observation of angiogenesis after coculture with angiogenic cells in vitro.

Yes, we evaluated the angiogenic potential of all the different ECM (soft tissues as well as hard bone tissues) but we did not evaluate the angiogenesis. As already mentioned above, thank you for your interesting comment. Indeed, it could add a benefit to co-culture PMSC with endothelial cells and evaluate some markers of angiogenicity and it is in our perspectives for a future work. We added this perspective in our discussion.

4) The author aims to create a novel osteogenic membrane used in critical bone defects to enhance the integration of massive bone graft . However, the manuscript does not reflect the advantages of its application. It is recommended to establish animal models for better verification.

Of course, you are totally right, as explained in the end of our discussion : “This new question raising from this in vitro study could be only addressed using in vivo models. As already developed in our institution [70], a femoral non-union pig model will assess the new osteogenic membrane in stringent conditions of a CSBD, typical of the clinical situation. This model will be representative of what orthopedic surgeons face in patients and will allow, in case of success, to consider the first clinical trials of this innovative approach.

It is a very interesting perspective but following the ethical steps of a fundamental research, we need to explore first in vitro experiments. These ones need to be read, approved and validated by peers before considering in vivo models. This way of working also allows to reduce the unnecessary animal experimentation and to respect the 3R concept (Reduce – Replace – Refine).

5) The English expression is not very clear, it is suggested to modify the language.

The entire manuscript was check again by at least 2 different authors which one of them are very fluent in English: he uses it as usual language.

We are very grateful for your comments and queries, which were quite thoughtful and in-depth for us. Therefore, we have adapted the manuscript accordingly, thereby improving the paper’s content, as requested. We modified the main text and manuscript by adding the previously explained modifications marked up using the “Track Changes” function as requested.

Once again, thank you so much for allowing us to clarify the open issues.

Sincerely Yours,

The authors.

Reviewer 4 Report

The goal of this study is clear: to create a novel osteogenic membrane to enhance the integration of massive bone allograft. The paper examines a variety of concepts and has organized the content in a well-organized, easy to follow format. All of the experiments have been rigorously performed with a thorough description of materials and methods, and the wide variety of assays employed strongly enforces the researcher’s findings. The figures are also highly informative and contain a lot of data, but the figure legends should be bolstered in order to make it easier for readers to interpret the data.

comments:

Figure 1 Legend: Although it is in the methods, the names of the assays performed should also be included in the figure legends for quick reference. 

Figure 3 Legend: For the quantitative analysis, include the names of the assay performed. Also explain what “Ratio D77/D1” is showing, I understand that it is part of the PMSC growth chart but it wasn’t clear upon first glance. I assume the PMSC Viability is showing the results of the Live/Dead assay, but that needs to be more explicit. 

Figure 4: Include the name of the assay that was used to obtain this data. If the same assay was used for both A) and B), it seems like it’s the same data but with different comparisons/visualizations. If this is correct, this should be made more clear to the reader. 

Tables 2&3: The legends assume the reader understands the statistical tests and what the various values mean, but a brief explanation of what the tests are determining and what the values mean (for example, what determines H) would make the tables easier to interpret. 

Figure 5: List the assays that were used in generating the quantitative data. As in Figure 4, clarify if A and B are showing different visualizations/comparisons of the same data.

There are a few typos throughout the paper, so check for spelling and grammar.

Author Response

Dear Reviewer,

On behalf of all the authors, I would like to thank you for the time and effort you dedicated to reviewing our manuscript titled “A new osteogenic membrane to enhance bone healing : At the crossroads between the periosteum, the induced membrane and the diamond concept” as well as for your pertinent comments and suggestions. This in-depth criticism has enabled us to significantly improve our manuscript. Please find below our point by point answers to your comments. Your initial feedback, copy-pasted in grey italic, is directly followed by the corresponding answer, written in normal black text.

Reviewer #4

Yes        Can be improved            Must be improved          Not applicable

Does the introduction provide sufficient background and include all relevant references?

              (x)          ( )          ( )           ( )

Are all the cited references relevant to the research?

              (x)          ( )          ( )           ( )

Is the research design appropriate?

              (x)          ( )          ( )           ( )

Are the methods adequately described?

              (x)          ( )          ( )           ( )

Are the results clearly presented?

              (x)          ( )          ( )           ( )

Are the conclusions supported by the results?

              (x)          ( )          ( )           ( )

The goal of this study is clear: to create a novel osteogenic membrane to enhance the integration of massive bone allograft. The paper examines a variety of concepts and has organized the content in a well-organized, easy to follow format. All of the experiments have been rigorously performed with a thorough description of materials and methods, and the wide variety of assays employed strongly enforces the researcher’s findings. The figures are also highly informative and contain a lot of data, but the figure legends should be bolstered in order to make it easier for readers to interpret the data.

  1. Figure 1 Legend: Although it is in the methods, the names of the assays performed should also be included in the figure legends for quick reference. 

The names of the assays were added in the legend.

  1. Figure 3 Legend: For the quantitative analysis, include the names of the assay performed. Also explain what “Ratio D77/D1” is showing, I understand that it is part of the PMSC growth chart but it wasn’t clear upon first glance. I assume the PMSC Viability is showing the results of the Live/Dead assay, but that needs to be more explicit.

The names of the assays and some explanations were provided in the legend.

  1. Figure 4: Include the name of the assay that was used to obtain this data. If the same assay was used for both A) and B), it seems like it’s the same data but with different comparisons/visualizations. If this is correct, this should be made more clear to the reader. 

It also has been added.

  1. Tables 2&3: The legends assume the reader understands the statistical tests and what the various values mean, but a brief explanation of what the tests are determining and what the values mean (for example, what determines H) would make the tables easier to interpret.

An explanation about the goal of the tests was added in both legends. However, the H value was already mentioned so as the test statistic of the Kruskal-Wallis test, i.e. the result/the value of the test/calculation.

  1. Figure 5: List the assays that were used in generating the quantitative data. As in Figure 4, clarify if A and B are showing different visualizations/comparisons of the same data.

It also has been added.

  1. There are a few typos throughout the paper, so check for spelling and grammar.

The entire manuscript was check again by at least 2 different authors.

We are very grateful for your comments and queries, which were quite thoughtful and in-depth for us. Therefore, we have adapted the manuscript accordingly, thereby improving the paper’s content, as requested. We modified the main text and manuscript by adding the previously explained modifications marked up using the “Track Changes” function as requested.

Once again, thank you so much for allowing us to clarify the open issues.

Sincerely Yours,

The authors.

Round 2

Reviewer 3 Report

The study aims to create a novel osteogenic membrane to enhance the integration of massive bone allograft. And it is the highlight of this study to explore an osteogenic membrane made of a decellularized collagen matrix from human fascia lata and seeded with periosteal mesenchymal stem cells (PMSC). Considering that the authors have carefully answered the questions previously raised and made targeted revisions, this revised manuscript can be accepted for publication. However, we suggest that the authors still need to make some modifications to the following problem.

1.     Since no relavant experiments have been conduted to demonstrate the problem of "Do MSC have to be used/transplanted already differentiated or not”, it's better to revise this description in the part of Introduction.

Author Response

Dear Reviewer,

On behalf of all the authors, I would like to thank you again for the time you dedicated to reviewing our manuscript. Your feedback, copy-pasted in grey italic, is directly followed by the corresponding answer, written in normal black text.

The study aims to create a novel osteogenic membrane to enhance the integration of massive bone allograft. And it is the highlight of this study to explore an osteogenic membrane made of a decellularized collagen matrix from human fascia lata and seeded with periosteal mesenchymal stem cells (PMSC). Considering that the authors have carefully answered the questions previously raised and made targeted revisions, this revised manuscript can be accepted for publication. However, we suggest that the authors still need to make some modifications to the following problem.

  1. Since no relavant experiments have been conduted to demonstrate the problem of "Do MSC have to be used/transplanted already differentiated or not”, it's better to revise this description in the part of Introduction.

This question has been clarified in the introduction. We modified the manuscript by adding the modifications marked up using the “Track Changes” function as requested. We are very grateful for your comments and queries allowing us to clarify the open issues.

Sincerely Yours,

The authors.